# Metronomic Chemotherapy Modulates Clonal Interactions to Prevent Drug Resistance in Non-Small Cell Lung Cancer

**DOI:** 10.3390/cancers13092239

**Published:** 2021-05-07

**Authors:** Maryna Bondarenko, Marion Le Grand, Yuval Shaked, Ziv Raviv, Guillemette Chapuisat, Cécile Carrère, Marie-Pierre Montero, Mailys Rossi, Eddy Pasquier, Manon Carré, Nicolas André

**Affiliations:** 1Centre de Recherche en Cancérologie de Marseille, Aix-Marseille Université, Inserm, CNRS, Institut Paoli Calmettes, 13273 Marseille, France; bondarenko.m@chu-nice.fr (M.B.); marion.le-grand@inserm.fr (M.L.G.); Marie-pierre.MONTERO@univ-amu.fr (M.-P.M.); mailys.ROSSI@etu.univ-amu.fr (M.R.); eddy.pasquier@inserm.fr (E.P.); 2Assistance Publique-Hopitaux de Marseille (AP-HM), Timone Hospital, 13385 Marseille, France; 3Cell Biology and Cancer Science, Rappaport Faculty of Medicine, Technion-Israel Institute of Technology, Haifa 3525433, Israel; yshaked@technion.ac.il (Y.S.); zraviv@technion.ac.il (Z.R.); 4Metronomics Global Health Initiative, 13385 Marseille, France; 5Centrale Marseille, CNRS, Aix Marseille Université, 13013 Marseille, France; guillemette.chapuisat@univ-amu.fr; 6Institut Denis Poisson, Université d’Orléans, CNRS, 45100 Orléans, France; cecile.carrere@univ-orleans.fr; 7Service d’Hématologie & Oncologie Pédiatrique, AP-HM, 13385 Marseille, France

**Keywords:** metronomic chemotherapy, intratumor heterogeneity, mathematical modeling, non-small cell lung cancer, glycolytic activity, lactate dehydrogenase

## Abstract

**Simple Summary:**

Inside a tumor mass, drug resistant and sensitive cell populations co-exist, leading to a therapeutic challenge in oncology. In this study, we developed 2D and 3D co-culture systems and mathematical modeling to better understand how these different populations impact each other. We demonstrated that drug-sensitive cell populations inhibit the growth of drug-resistant populations. Mathematical modeling predicted that metronomic schedules, using chronic administration of drugs at low doses, could better control intratumor cell dynamics. We validated our in silico data in 3D and in vivo models. Finally, we demonstrated that metabolic cell activity of drug-sensitive cells could play a key role in controlling the proliferation of drug-resistant cells. Altogether, our study reports a new mechanism of action of metronomic therapy and paves the way for using it to better control drug resistance.

**Abstract:**

Despite recent advances in deciphering cancer drug resistance mechanisms, relapse is a widely observed phenomenon in advanced cancers, mainly due to intratumor clonal heterogeneity. How tumor clones progress and impact each other remains elusive. In this study, we developed 2D and 3D non-small cell lung cancer co-culture systems and defined a phenomenological mathematical model to better understand clone dynamics. Our results demonstrated that the drug-sensitive clones inhibit the proliferation of the drug-resistant ones under untreated conditions. Model predictions and their experimental in vitro and in vivo validations indicated that a metronomic schedule leads to a better regulation of tumor cell heterogeneity over time than a maximum-tolerated dose schedule, while achieving control of tumor progression. We finally showed that drug-sensitive and -resistant clones exhibited different metabolic statuses that could be involved in controlling the intratumor heterogeneity dynamics. Our data suggested that the glycolytic activity of drug-sensitive clones could play a major role in inhibiting the drug-resistant clone proliferation. Altogether, these computational and experimental approaches provide foundations for using metronomic therapy to control drug-sensitive and -resistant clone balance and highlight the potential of targeting cell metabolism to manage intratumor heterogeneity.

## 1. Introduction

The advent of genomic medicine has increased our appreciation of intratumoral heterogeneity, which refers to clonal diversity among the tumor cells of a single patient. Even after malignant transformation, a cancer remains dynamic. This ongoing evolution generates a heterogeneous tumor mass consisting of cancer cells harboring distinct molecular signatures [1]. Intratumor heterogeneity has emerged as a key mechanism that underlies therapeutic resistance and constitutes a genuine challenge for oncologists [2,3]. Drug administration has, in itself, direct phenotypic consequences on tumor behavior and the emergence of resistant clones [4]. Selective pressures exerted by anti-cancer drugs can lead to the expansion of resistant clones that either existed before the onset of therapy or that emerged as a result of treatment administration [2,3]. Deciphering how tumor clones progress and impact each other could provide valuable biological insights and unveil vulnerabilities that could be therapeutically targeted.

Most research efforts are still focused on treatments that maximally kill tumor cells while minimizing toxicity to the host. Conventional chemotherapeutic drugs are currently predominantly administrated at or near the maximum tolerated dose (MTD), which gives the largest possible amount of the drug at the beginning of the cycle and then lets the patient recover from toxicities. However, it has been demonstrated that a cancer clone competition occurs within each individual tumor, with the drug-sensitive subpopulation of tumor cells proliferating at the expense of the resistant phenotype cells [5]. In this context, the traditional MTD protocol leads to the elimination of chemosensitive clones, making way for the emergence of resistant clones [6,7]. Modulating the dose and frequency of chemotherapy administrations in order to manage a constant tumor volume could bring substantial benefits. The concept of ‘adaptive therapy’, first pioneered by Gatenby, where drug dose is modulated in response to the underling evolutionary dynamics, has showed promising results in controlling intratumoral competition in ovarian cancers [5,8]. Another treatment schedule called metronomic chemotherapy (MC), defined as the chronic administration of chemotherapeutic agents at relatively low, minimally toxic doses and with no prolonged drug-free breaks [9,10], is slowly becoming appreciated. Accumulating clinical evidence indicates that MC can offer equal, if not better, antitumor efficacy than traditional MTD regimens in different types of cancers, including breast, prostate and lung cancers [9,11,12]. The molecular mechanisms underlying the MC effects rely on anti-angiogenic activity, as well as restoration of the anticancer immune response and induction of tumor dormancy [12,13]. However, whether the enhanced control of cancer progression in MC treated patients could be mediated through an effect on intratumor heterogeneity balance remains unknown.

In this study, we developed 2D and 3D heterogeneous non-small cell lung cancer (NSCLC) co-culture models and a phenomenological mathematical model to better understand intratumor heterogeneity. MC was predicted to better control tumor volume than the MTD protocol, without the emergence of resistant clones. We also revealed for the first time that metabolic activity of drug-sensitive clones could play a key role in controlling the proliferation of drug-resistant clones.

## 2. Results

### 2.1. The Co-Culture System Demonstrates a Balance between Drug-Sensitive and Drug-Resistance Clones

Genomic diversity within single tumors has long been recognized. It is now well-known that drug-resistant clones co-exist with drug-sensitive clones in untreated tumors [14]. In this study, we developed a simple co-culture method for adherent NSCLC cells to mimic intratumor heterogeneity in vitro, which allowed for long-term studies (up to 3 weeks). Using GFP and DsRed stably expressing cells, we were able to analyze over time the tumor subpopulations dynamics within these two-dimensional (2D) heterogeneous co-cultures (Appendix A). A549/EpoB40-GFP NSCLC cells, which are resistant to cisplatin and patupilone (Appendix A), and the parental drug-sensitive A549-DsRed NSCLC cells were used at the initial co-culture seeding ratio of 20/80% respectively. As shown in Figure 1a,b, drug-sensitive A549 clones continuously proliferated in the whole well, with an invasion of the drug-resistant cells’ initial seeding zone from day 6. In the meantime, the drug-resistant A549/EpoB40 clones did not expand and fluorescence analysis confirmed the decrease in the drug-resistant subpopulation, which was progressively substituted by the drug-sensitive clones (Figure 1a,b). Similar results were obtained with other 2D co-culture models, including an etoposide-resistant A549/VP16 NSCLC model and an oxaliplatin-resistant HT29/Rox1 colorectal cancer model (Appendix A). As drug-resistant cells are described as being less fit than the drug-sensitive cells due to the phenotypic cost of resistance [7], cell growth kinetic assays were performed in drug-sensitive A549 and drug-resistant A549/EpoB40 cells. Our results demonstrated that the doubling time difference between cell clones was not significant (Appendix A). Moreover, to definitively ascertain the suppressive effect of the drug-sensitive clones over the proliferation of the drug-resistant ones, different cell clone ratios were used. As shown in Figure 1c, in a homogeneous co-culture, the A549/EpoB40 cell proliferation was constant over time until reaching well confluency. When drug-resistant A549/EpoB40 cells represented 40 or 80% of the initial co-culture seeding, a suppressive effect of the drug-sensitive parental cell clones on the A549/EpoB40 cell proliferation was still observed. Our data thus confirm that the inhibition of drug-resistant clone growth resulted from their co-culture interaction with the drug-sensitive clones. We then developed a NSCLC 3D spheroid co-culture model that better recapitulates cell–cell interactions in carcinomas. Fluorescence signals were converted to a number of drug-resistant cells using a calibration curve (Appendix A). Consistent with the 2D co-culture results, fluorescence analysis demonstrated a stabilization of drug-resistant clone growth within hetero-spheroids, while A549/EpoB40 cells continuously proliferated within homo-spheroids (Figure 1d). Altogether, these results validate our 2D and 3D co-culture models, showing a dominant role of the drug-sensitive clones over the drug-resistant ones in a NSCLC model under untreated conditions.

### 2.2. Mathematical Modeling Predicts that Metronomic Treatment Could Better Manage Intratumor Heterogeneity than the MTD Schedule

In order to better understand the influence of drug-sensitive clones over the drug-resistant ones, we computed a mathematical model, which is presented in details in Appendix A and in our previous mathematical article [15]. It took the following biological hypotheses into account:In the well, there may be two types of cells, denoted respectively as drug-sensitive and drug-resistant cells.In the absence of the other cell type, drug-sensitive and -resistant cells can grow freely until the well becomes confluent.All cells from both types are competing to colonize the available space in the well.The presence of drug-sensitive cells in the well may have a suppressive role on the drug-resistant clone proliferation.

Our model was calibrated upon our experimental data carried out with A549 and A549/EpoB40 cells. Our first aim was to study whether a suppressive action of drug-sensitive clones over the resistant ones was actually required in order to explain our results presented above. First, we assumed that both cell clones were growing without a suppressive effect of the drug-sensitive cells (β = 0 in the model, Appendix A). A mathematical analysis of the equations showed that, whatever the value of the parameters may have been, the number of drug-resistant cells increased over time during all the experiments, maintaining a co-culture ratio that could not be neglected, as shown in Figure 2a (Appendix A). We then calibrated a suppressive effect of the drug-sensitive cells over the drug-resistant ones (β > 0 in the model). In this case, an asymptotic analysis of the equations suggests that the drug-resistant cell proliferation decreased over time until it became negligeable, while the drug-sensitive cells continuously grew (Figure 2a and Appendix A). These results demonstrated that our mathematical model fits better with our experimental data when a suppressive effect is taken into account.

In order to understand the influence of the suppressive action of sensitive cells upon the efficiency of a treatment, we then introduced the effect of chemotherapy in our mathematical model (Appendix A). The biological hypotheses considered were the following:The chemotherapeutic agent only acts on drug-sensitive cells.A delay between the injection of the chemotherapeutic agent and its impact on cell functions is taken into account.The chemotherapeutic agent concentration is assumed constant in the absence of an experimenter intervention.

Taking into account the suppressive effect of drug-sensitive clones over the drug-resistant ones, simulations of different dosing regimens were performed. Our model predicted that following a MTD treatment, the drug-sensitive cell proliferation was strongly inhibited, while the drug-resistant cell proliferation resumed after a certain delay until the well was confluent (Figure 2b). We then changed the schedule of administration using a metronomic protocol (MC). A long-term control of drug-sensitive cell proliferation was obtained, with no drug-resistant cell proliferation (Figure 2b). The analysis of the mathematical model therefore suggests that a metronomic schedule should be more efficient than MTD-based conventional chemotherapy to maintain a heterogeneous tumor that is as small as possible. 

### 2.3. In Vitro and In Vivo Biological Validations of Mathematical Model Predictions

To validate our mathematical model, we evaluated the antitumor efficacy of two treatment regimens in the 2D heterogeneous co-culture system: a MTD-like treatment schedule based on the IC_50_ values of the drug-sensitive cells (5 nM of patupilone, once a week for 24 h) and a MC-like treatment schedule based on protracted low-dose drug administration (0.5 nM of patupilone, five times a week). Both treatment schedules led to a global decrease of tumor proliferation rate in two different ways. Indeed, the MTD-like treatment schedule was initially effective in decreasing proliferation of the drug-sensitive A549-DsRed cells (Figure 3a, red population), while the MC-like treatment schedule only resulted in limiting the drug-sensitive A549-DsRed proliferation (Figure 3b, red population). Meanwhile, as predicted by our mathematical model, the drug-resistant A549/EpoB40-GFP clones started to grow from day 5 under the MTD-like treatment schedule, and proliferated until the surface of the well was fully confluent (Figure 3a, green population). In sharp contrast, the MC-like treatment schedule was able to indirectly suppress the growth of the drug-resistant A549/EpoB40-GFP clones by controlling the growth of A549 sensitive clones (Figure 3b, green population). Collectively, these data suggest that the MC could better control tumor growth than the MTD-like treatment schedule by maintaining the dynamic of intratumor heterogeneity and validate our mathematical model. Furthermore, we were also able to predict a specific threshold of the chemotherapeutic agent concentration, called C*_threshold_*, which keep a small proportion of drug-sensitive cells for maintaining a repressive effect on the drug-resistant cell proliferation (Appendix A). Using our experimental data, our model has been calibrated, and our results indicated that the C*_threshold_* was at 0.584 nM (Figure 3c). 

We then confirmed our 2D heterogeneous co-culture data and mathematical simulations in a 3D co-culture model. Our results demonstrated that both treatments significantly increased the A549/EpoB40 drug-resistant population in comparison to control (Figure 4a). However, the proliferation of the drug-resistant clones was partially repressed under the MC-like treatment in comparison to the MTD-like treatment (Figure 4a). Indeed, while the drug-resistant cell proliferation increased by a factor of 2.1 under the MC-like treatment, a three-fold increase was observed under MTD-like treatment (Figure 4a). Moreover, the MC-like treatment effect on controlling the proliferation of the drug-resistant population was also validated in the 3D HT29/Rox1 colorectal cancer model (Figure 4b). While no difference was observed between the non-treated condition and the metronomic schedule, the drug-resistant cell proliferation significantly increased by a factor of 2.7 under the MTD schedule (Figure 4b). To validate our results in an in vivo model, a mosaic tumor-bearing mice model was used. To first ensure the constant cell heterogeneity of our model over time in the absence of treatment, we analyzed ex vivo the tumor cell composition at three different time points: when tumor size reached 100 and 200 mm^3^ and at the study end point. As shown in Figure 4c, the tumor cell composition did not change over time with a mixture of drug-sensitive A549-mtDsRed and drug-resistant A549/EpoB40-GFP. MTD-like (paclitaxel at 25 mg/kg, once every 3 weeks) or MC-like (paclitaxel 1.2 mg/kg, daily) treatment schedules were then administered. Both treatments had a significant repressive effect on tumor growth (–67 +/−8% and –48 +/−4% for MTD- and MC-like treatment schedules, respectively, in comparison to the control; Figure 4d). Although there was no significant difference between the MTD- and the MC-like treatments on the tumor growth at the time of the study end point, our data demonstrated that, depending on the schedule administered, the drug-resistant cell population inside the tumor mass was modified. Indeed, following the MC-like treatment, the A549/EpoB40 drug-resistant cells represented 22 +/−4% of the tumor mass, while the MTD-like treated tumor cells were composed by 58 +/−5% of the A549/EpoB40 drug-resistant cells (Figure 4e). Even if a significant decrease in the drug-resistant population was also shown following the MC-like treatment in comparison to the vehicle-treated (22 +/−4% vs. 38 +/−2.5%, *p* < 0.05, Figure 4e), the drug-sensitive cell population was still predominant inside the tumor in both cases. Collectively, these results validate our mathematical model predictions and indicate that MC leads to a better control of NSCLC cell heterogeneity than the MTD schedule over time, while achieving control of the global tumor volume.

### 2.4. Drug-Sensitive Clones Control the Proliferation of Drug-Resistant Clones Through Indirect Cell-Cell Interaction

It is now well-known that indirect cell–cell communication could play a key role in cancer [16]. To investigate whether the drug-sensitive NSCLC clones could exert their suppressive effect on the drug-resistant ones without direct cell interaction, we used Transwell^®^ systems of heterogeneous co-cultures. Our results demonstrated that in a homogeneous co-culture, A549/EpoB40 cells continuously grew over time (Figure 5a). However, drug-resistant cell proliferation could be inhibited by indirect interaction with the drug-sensitive cells (Figure 5a). This repressive effect was not specific to A549 cells as similar results were obtained with H1650, H1975 and HCC827 NSCLC cell lines (Figure 5a). We further analyzed the effect of conditioned media using cell-free supernatants on the drug-resistant A549/EpoB40 cell growth by real-time impedance measurement. Our results showed a decrease in the proliferation of drug-resistant A549/EpoB40 cells treated with drug-sensitive A549 supernatants in comparison to treatment with the drug-resistant A549/EpoB40 supernatants (Figure 5b, grey-shaded area). This was reflected by a significant drop in the A549/EpoB40 cell growth curve slope treated with the A549 supernatant condition of 56 +/−3% (*p* < 0.01), demonstrating an indirect interaction between the two cell clones (Figure 5b,c). Moreover, the discontinuation of exposure to drug-sensitive A549 supernatants led to a recovery of the drug-resistant A549/epoB40 cell proliferation (Figure 5b, white-shaded area), as indicated by the same slopes for both cell populations (Figure 5c). As it has recently become apparent that secreted extracellular vesicles (EVs) are proficient intercellular communication mediators [17], we separated vesicles from the soluble fraction of drug-sensitive A549 cell-free supernatants by sequential centrifugations and analyzed their effect on drug-resistant A549/EpoB40 cell growth. The vesicles we extracted were exosomes based on their diameter, ranging from 50 to 200 nm, as previously described [18]. Our results showed that treatment with a soluble fraction from drug-sensitive A549 cells significantly decreased the drug-resistant A549/EpoB40 cell viability over time as did A549 cell-free supernatant treatment, in agreement with the impedance results (Figure 5d). In comparison, EVs fraction did not significantly impact the drug-resistant A549/EpoB40 cell viability (Figure 5d). Collectively, our data indicate that drug-sensitive A549 clones can control the proliferation of drug-resistant A549/EpoB40 clones through an indirect cell–cell communication, independently of exosome secretion.

### 2.5. The Metabolic Activity of Drug-Sensitive Clones Is a Key Factor to Maintain Their Repression on Drug-Resistant Clones

It is now established that metabolic reprogramming happens in tumor tissue, resulting in cancer cell phenotypes with different metabolic activities, leading to adaptive/acquired resistance to anti-tumor therapy [19]. To investigate whether the metabolic status of drug-sensitive A549 and drug-resistant A549/EpoB40 clones could be different, we performed real-time measurements of the oxygen consumption rate (OCR) and extracellular acidification rate (ECAR), reflecting the mitochondrial and glycolytic activities, respectively. As illustrated in Figure 6a, drug-resistant A549/EpoB40 cells had a higher overall mitochondrial metabolic efficiency than drug-sensitive A549 cells. Indeed, basal OCR increased by 85 +/− 7% (*p* < 0.001) in comparison to the drug-sensitive A549 cells. Moreover, maximal mitochondrial respiration, induced by FCCP, increased by 101 +/− 16% (*p* < 0.001) in comparison to the drug-sensitive A549 cells. Meanwhile, our results showed that drug-resistant A549/EpoB40 cells were not able to increase their glycolytic capacities following oligomycin administration, which suppresses mitochondrial energy production (Figure 6b). This demonstrates that drug-resistant cells are more dependent on mitochondrial respiration than drug-sensitive cells, which mostly rely on glycolytic activities. As this type of altered energetic metabolism can result in the secretion of specific metabolites that can have a significant impact on tumor progression, dissemination and drug response [19], we next measured extracellular glucose and lactate concentration in cell-free supernatants. Our results showed that drug-sensitive A549 cells exhibited higher glucose consumption and lactate production than drug-resistant A549/EpoB40 cells. Indeed, at 48 h, relative glucose and lactate concentrations were 83 +/− 4% lower and 93 +/− 15% higher, respectively, in the drug-sensitive A549 cell-free supernatant in comparison to the drug-resistant A549/EpoB40 cell-free supernatant (Figure 6c,d). As our results suggest an important role of the glycolytic pathway in the drug-sensitive A549 cells, we focused on the potential role of LDHA (Lactate dehydrogenase A) being the primary metabolic enzyme that converts pyruvate to lactate in cancer cells [20] and a hallmark of aggressive cancers [21,22]. We used FX11, a pharmacological inhibitor of LDHA, at a non-cytotoxic concentration of 3 µM (Appendix A). Our results first showed that pre-treatment of drug-sensitive cells with FX11 was able to significantly reverse the suppressive effect of cell-free supernatants from A549, H1975 or HCC825 cells on the drug-resistant A549/EpoB40 cell proliferation (Figure 6e, *p* < 0.01). Our data also showed that supernatants from FX11 pre-treated drug-sensitive A549 cells restored the proliferative rate of drug-resistant A549/EpoB40 cells (Figure 6f, Appendix A). Altogether, our results suggest that the glycolytic activity of drug-sensitive A549 clones could play a major role in drug-resistant A549/EpoB40 clone proliferation.

## 3. Discussion

Lung cancer is the leading cause of cancer-related death worldwide, with non-small-cell lung cancer (NSCLC) being the most common type [23]. Despite the emergence of targeted therapies and immunotherapies, chemotherapy is still a key component of NSCLC treatment [24]. Studies evaluating intratumor heterogeneity and cancer genome evolution in lung cancer patient cohorts have revealed that intratumor heterogeneity is a key factor contributing to drug resistance, therapeutic failure and lethal outcome of lung cancer [25,26]. The lack of reliable experimental model systems that recapitulate clonal evolution is a major challenge to better understand cancer cell heterogeneity. Here, we developed simple heterogeneous NSCLC co-culture models with chemotherapeutic drug-sensitive and -resistant clones to study the tumor clone dynamic. Our data demonstrated a competition between the drug-sensitive and drug-resistant clones, where the former exerted a suppressive effect on the proliferation of the latter in the absence of treatment. Moreover, our in silico, in vitro and in vivo data demonstrated for the first time that MC leads to better control of NSCLC cell heterogeneity than the standard-of-care MTD-like treatment schedule. By demonstrating the role of MC in controlling intratumor heterogeneity, our study indicates that MC might represent a better therapeutic schedule in NSCLC. During the last decade, numerous clinical trials have been performed to explore MC in first- and second-line treatment, and to investigate the maintenance of treatment for metastatic NSCLC [27,28,29,30]. Due to lower toxicity, higher tolerability and acceptable safety, MC could be a better option for patients with NSCLC cancer [31]. In our study, the same impact of MTD and MC schedules on in vivo NSCLC tumor growth was reported. This result might be explained by our short-time study, and further in vivo analyses with a longer treatment period are still required. Moreover, to date, it is still unclear whether the cumulative drug concentration could play a key role in the mechanism of action of MC. However, it is more likely that specific signaling pathways and different cell types would be targeted depending on the temporal administration protocol used (continuous rather than intermittent) [9,32]. 

MC is an intrinsic multi-targeted therapy believed to mainly act on the tumoral micro-environment through modulation of angiogenesis. However, despite being used for several decades, the mechanisms of MC still need to be fully determined. Among the additional anticancer mechanisms that have been unveiled, stimulation of an antitumor immune response has now been well-established. Indeed, low doses of cytotoxic drugs such as cyclophosphamide or temozolomide can switch the immunological balance from immunosuppression to immune stimulation [33,34]. Moreover, it has been demonstrated that MC can directly target tumor cells [32]. For instance, Orlandi et al. demonstrated that metronomic vinorelbine can inhibit ERK and AKT signaling pathways in NSCLC, while several studies have demonstrated the impact of MC in directly targeting cancer stem cells [35,36,37]. Here, our results demonstrated that MC better manages the preservation of intratumor heterogeneity than the MTD schedule, thus avoiding the selection of drug-resistant clones. As most of our experiments were performed in vitro, our study reinforces the existence of a direct effect of MC on cancer cells [32]. Moreover, our in vivo data also showed that the MC schedule led to a decrease in the drug-resistant population in comparison to the vehicle condition, while maintaining a dominant proportion of drug-sensitive cells in both cases as expected by our mathematical model. These results could be explained by the direct effect of MC on cancer cells for its global anti-tumor effect. Even if our findings contribute to a better understanding of MC’s mechanism of action by revealing a new effect on clonal heterogeneity, our study indicated the importance of better understanding the different drug mechanisms of action, depending on the schedule administered. Furthermore, more investigations will be needed to incorporate tumor microenvironment cells in our models, such as cancer-associated stromal cells and immune cells.

With many effective therapies now available, it has become more and more difficult to determine the optimal combinatorial treatments and schedules, resulting in an unprecedented number of failed clinical trials over the past few years [38,39]. Mathematical modeling has rapidly evolved and could provide relevant tools in oncology [40,41]. Indeed, preclinical and clinical trials in which mathematical models have been used to estimate the mechanism(s) of treatment failure and explore alternative strategies with new drugs and schedules showed their potential to improve therapeutic efficacy [42,43,44,45]. Here, our in silico prediction revealed that MC does better than the MTD schedule to manage NSCLC intratumor heterogeneity. This was furthermore confirmed in a 3D spheroid and mouse xenografted model, supporting the interest in mathematical models to predict optimal treatments. Moreover, our models were also able to predict the clonal competition between drug-sensitive and -resistant clones. The clonal intratumor dynamic has led several researchers to develop modeling and simulation tools for describing biological and pharmacodynamics processes and aiming at managing resistance phenomena [5,46]. More work must be done in the future to combine both types of models (i.e., cancer clone dynamics with treatment schedule models) and design optimal treatment strategies to manage intratumor heterogeneity. 

In evolutionary cancer treatment, a key component of the Darwinian dynamics is the cost of resistance. Cancer cells must alter their phenotype to become resistant [47]. To support the synthesis of biomass components and to generate energy required for cellular growth, cancer cells have to reshape the regulatory and functional properties of their metabolic networks [48]. Cancer cells preferably use aerobic glycolysis to generate energy, which has been recognized as a hallmark of cancer [49]. In our models, we demonstrated that drug-sensitive NSCLC clones mostly relied on aerobic glycolysis, while drug-resistant clones relied on OXPHOS activity. This observation is consistent with other studies reporting resistant cells as being more addicted to mitochondrial functions in different cancer models, including melanoma, leukemia and pancreatic cancers [50,51,52]. Our results also revealed for the first time that the metabolic activity of drug-sensitive clones could play a key role in controlling the proliferation of drug-resistant ones. We highlighted that drug-sensitive cells exhibited higher glucose consumption and lactate production than drug-resistant cells. Lactate dehydrogenase A (LDHA) is a key glycolytic enzyme, a hallmark of aggressive cancers and is believed to be the major enzyme responsible for pyruvate to lactate conversion [20]. Thus, inhibition of this key metabolic enzyme has been demonstrated as a promising strategy for cancer treatment [53,54]. Our data, however, suggested that LDHA activity could play a major role in the regulation of drug-resistant clone proliferation. Inhibiting its activity might then lead to the rise of resistant clones. To support this hypothesis, more work is required to better understand the role of LDHA in the cancer cell clone interactions.

## 4. Materials and Methods 

### 4.1. Cell Lines

The NSCLC cell lines used were A549 (RRID:CVCL_0023), A549/EpoB40 (epothilone B highly resistant cells derived from A549, kindly given by Horwitz SB, AECOM, NY, USA), A549/VP16 (etoposide-resistant cells derived from A549, kindly given by Morjani H, Reims, France), H1975 (RRID:CVCL_1511), H1650 (RRID:CVCL_1483) and HCC827 (RRID:CVCL_2063) cell lines, and the human colon adenocarcinoma cell lines were HT29 (RRID:CVCL_0320) and HT29/Rox1 (oxaliplatin-resistant cells derived from HT29, kindly given by Leloup L, INP, Marseille, France). All resistant cell lines were generated by repeated treatments of cells for an extended period of time. The human NSCLC cell lines were grown in an RPMI-1640 medium (Gibco, Thermo Fisher Scientific, Villebon-Sur-Yvette, France), and the human colon adenocarcinoma cell lines were cultured in a DMEM medium (Gibco, Thermo Fisher Scientific, France). All the media were supplemented with 10% fetal bovine serum (FBS), 1% glutamine and 1% penicillin streptomycin (Gibco, Thermo Fisher Scientific, France). Patupilone 40 nM was added to the RPMI-1640 medium for routine A549/EpoB40 cell culture and for cell adhesion. NSCLC cell lines were transfected by the mitochondrion-targeted DsRed (mtDsRed, Takara Bio, Saint-Germain-En-Laye, France) plasmid or the enhanced green fluorescent protein (EGFP) plasmid (pEGFP-C1, Clontech/Takara Bio Europe). Transfections were achieved using Lipofectamine 2000 (Invitrogen, Thermo Fisher Scientific, Villebon-Sur-Yvette, France) according to the protocol supplied by the manufacturer. Stable cell lines were obtained after geneticin selection (800 µg/mL; Thermo Fisher Scientific, France). Cell sorting (FACScan, BD Biosciences) was also performed systematically to obtain high quality fluorescence cells. Cells were maintained in culture at 37 °C with 5% CO_2_ and regularly screened to ensure the absence of mycoplasma contamination (MycoAlert, Lonza, Colmar, France).

### 4.2. Drugs and Reagents

Patupilone (epothilone B, Novartis, Rueil-Malmaison France), etoposide and cisplatin (Mylan, Canonsburg, PA, USA), oxaliplatin (Sanofi-Aventis, Raspail, France) and FX11 (cat. no. 427218; Merck Millipore, Lyon, France) were prepared in dimethylsulfoxide (DMSO, VWR) and were freshly diluted in the culture medium for experiments. The highest concentration of DMSO to which the cells were exposed was 0.001%.

### 4.3. Establishment of Homo- and Heterogeneous 2D Co-Culture Models

To yield experimental heterogeneous adherent cell co-cultures, 4.6 × 10^4^ drug-sensitive cells and 1 × 10^4^ drug-resistant cells were simultaneously seeded in a 6-well plate. Drug-sensitive A549 and drug-resistant A549/EpoB40 cells were first seeded in their specific culture medium. The two cell lines were initially separated by a silicon insert that was removed 24 h after seeding. The homogeneous 2D co-cultures, which served as controls, included either A549-DsRed with A549-GFP or A549/EpoB40-DsRed with A549/EpoB40-GFP, in the same proportions as in the heterogeneous structures. Treatments were administered in culture media, which were changed 5 days per week as follows: (i) the culture medium was changed 5 times a week in non-treated conditions and for cells exposed to MC regimens (0.5 nM patupilone); (ii) in the MTD conditions, cells were exposed to drug (5 nM patupilone) for 24h and incubated in daily-changed drug-free culture medium for the four other days. To analyze the sensitive and resistant cell growth over time, DsRed and GFP fluorescent signals were recorded with a well-scanning mode microplate reader (PHERAStar FS, BMG LABTECH, Champigny-Sur-Marne, France). Images were acquired by a Leica DM-IRBE microscope (Leica Microsystems, Wetzlar, Germany). Cell proliferation was expressed as a percentage of the fluorescence signal measured at day 0.

### 4.4. Establishment of Homo- and Heterogeneous 3D Co-Culture Models

To obtain heterogeneous co-culture spheroids in 96-well round bottom plates, drug-resistant cells were resuspended with drug-sensitive cells at the same ratio as the 2D co-culture models in a volume of 100 μL culture media, supplemented with 20% of methylcellulose (Sigma-Aldrich, Lyon, France). The control 3D structures (homogeneous co-cultures) included A549/EpoB40-DsRed with A549/EpoB40-GFP cells, in the same proportions as the heterogeneous structures. After 48h, spheroids were treated following MC (0.5 nM patupilone or 0.2 µM oxaliplatin) and MTD treatment schedules (5 nM patupilone or 2 µM oxaliplatin). Resistant cell growth within the 3D structures was analyzed daily by recording the fluorescent signals as described for 2D co-cultures. Each signal was then converted into a number of resistant cells present in the structure, via a range of calibration carried out previously with 3D spheroids made of 500 to 150,000 A549/EpoB40-DsRed cells. 

### 4.5. Cell Viability Assay

Cells were seeded in 96-well plates to be treated over 72 h with cytotoxic drugs. Cell survival was measured by using the colorimetric MTT assay (Sigma-Aldrich) as we previously performed [55]. 

### 4.6. Proliferation Assay

A549/EpoB40 cells were grown in 96-well plates for 24 h, and then, they were exposed daily to conditioned media from A549, H1975 or HCC825 cultures, previously treated or not-treated with FX11. Conditioned media from A549/EpoB40 cells were used as controls. Conditioned media were obtained from cells seeded in 6-well plates. Before treatment, cell-conditioned media were centrifuged for 30 min at 10,000× *g* at 4 °C. At different indicated time points, cells were fixed with 1% glutaraldehyde and stained with a 1% crystal-violet solution in 20% methanol. The stain was eluted with DMSO, and absorbance was measured at 600 nm with a Multiskan Ascent plate reader. 

### 4.7. Impedance Measurements

The label-free dynamic monitoring of cell proliferation and viability in real time was done by using the xCELLigence technology of the RTCA SP system (ACEA Biosciences, Ozyme, Saint-Cyr-L’Ecole, France). Cells were seeded in 96-well microplates (E-Plates). Attachment of the cells was followed every 5 min over 24h, and cell proliferation was then monitored daily every 15 min until cell confluency. Cell-sensor impedance was expressed as cell index values that were normalized using the RTCA Software 2.0. Briefly, the software automatously and uniformly transforms the cell index values in different wells at the base-time, defining as the first measurement after starting treatment as 1. This made the normalized cell index more comparable between wells. The resulting normalized cell index was actually the relative cell impedance, presented as the percentage of the value at the base-time.

### 4.8. Transwell Co-Culture System

A 24-Multiwell Insert System with 1 µm pore size was used (BD Falcon, BD Biosciences, Grenoble, France). Cells were seeded at 1200 cells in both upper and lower chambers to ensure sustained exponential growth for 10 days. Cell culture media were changed in both upper and lower chambers every day, in agreement with 2D and 3D co-culture experiments. Two experimental seeding conditions were tested: (1) A549-DsRed in the upper chamber and A549/EpoB40-GFP in the lower chamber; (2) A549/EpoB40-DsRed in the upper chamber and A549-GFP in the lower chamber. A549/EpoB40-DsRed and A549/EpoB40-GFP in the upper and lower chambers, respectively, served as controls. Fluorescence microscopy was used to ensure that no cell migrated from one compartment to the other. Similar experiments were also conducted with H1650, H1975 or HCC825 seeded in the upper chamber and with A549/EpoB40 cells seeded in the lower chamber. At day 3 and day 7, cell proliferation in the wells and the inserts were determined by using crystal violet staining, as described in the proliferation assay section.

### 4.9. Extracellular Vesicles Isolation

Extracellular vesicles (EVs) were isolated from the supernatant of A549 and A549/EpoB40 cells by differential centrifugations. The cell culture medium was centrifuged at 100,000× *g* for 3 h. Soluble fractions were filtered through a 0.22 µm sterile filter and then mixed with serum-free medium. Cells were seeded in a T25 flask and incubated overnight. Cell conditioned media were replaced by EVs-depleted cell culture medium at 24, 48 and 72 h. Cell supernatants were collected at different time points and centrifuged for 30 min at 10,000× *g* at 4 °C to eliminate cell debris (centrifuge CT15RE, VWR), then at 100,000× *g* for 3 h to eliminate larger vesicles (L7-55 Ultracentrifuge, Beckman Coulter, Brea, CA, USA). The pellet was washed with PBS and ultracentrifuged at 100,000× *g* 1 h at 4 °C to obtain small vesicles (TL-100 Ultracentrifuge, Beckman Coulter). The final pellet was resuspended in PBS for subsequent experiments. EVs were fixed with 2% uranyl acetate and were visualized by using transmission electron microscopy to quantify their sizes (JEOL 1220, Tokyo, Japan). To evaluate involvement of A549 total and fractionated conditioned media on A549/EpoB40 cell viability, A549 and A549/EpoB40 cells were seeded in 96-well plates (2,500 cells/well). At 24, 48 and 72 h, cells were treated with the total cell-free supernatant, EVs and the soluble fraction from A549 and A549/EpoB40 cells (the latter being used as a control). Cell viability was measured using an MTT assay at different time points, as described in the cell viability assay section.

### 4.10. Real-Time Metabolic Analysis

Multiparameter metabolic analysis of intact cells was performed in the Seahorse XF24 extracellular flux analyzer (Seahorse Bioscience, Billerica, MA, USA) as previously described [56]. Briefly, A549 and A549/EpoB40 cells were seeded in Seahorse XF24 plates (1500cells/well) and incubated overnight at 37 °C in 5% CO_2_. To normalize OCR and ECAR data to the cell number, A549 and A549/EpoB40 cells were simultaneously seeded in a second multi-well plate for 24 h and were then stained with crystal violet, as described in the proliferation assay section.

### 4.11. Glucose Consumption and Lactate Production Assay

YSI 2950 (Life Sciences, Hertfordshire, UK) was used to measure the total flux of glucose and lactate, as previously described [56]. At each time point, media were collected, centrifuged for 5 min at 1,200 rpm and kept at −20 °C before measurement. Metabolite concentrations were normalized to cell number as described for real-time metabolic analysis. 

### 4.12. In Vivo Studies

Eight week old male NOD SCID mice, generously obtained from the laboratory of Prof. Israel Vlodavsky (Rappaport Faculty of Medicine, Technion, Haifa, Israel), were injected subcutaneously to the right flank with A549 cells together with A549/EpoB40 cells, in a ratio of 7:3 (sensitive: resistant), respectively. Tumors were allowed to grow, and when they reached the size of 200 mm^3^, mice were randomly divided into three groups and treated with (i) MTD paclitaxel (25mg/kg, intraperitoneally, once every 3 weeks); (ii) MC paclitaxel (1.2 mg/kg, intra-peritoneally, daily) or (iii) vehicle control. Tumor growth was monitored twice a week using a caliper. Mice were sacrificed at day 26 post-treatment initiation. Tumors were then removed, and single cell suspensions were produced as previously described [57]. Vehicle tumors were also sampled at the size of 100 and 200 mm^3^. Cells were then incubated with an APC-conjugated anti-human HLA antibody (clone W6-32; cat. no. 311410; BioLegend, San Diego, CA, USA). Cells were subsequently analyzed by flow cytometry for GFP cells/DsRed cells composition of the tumor mass. Animal studies were performed in accordance with the Animal Care and Use Committee of the Technion-Israel Institute of Technology (Haifa, Israel–IL-089-07-2016).

### 4.13. Statistical Analysis

Each experiment was performed at least in triplicate. Data are presented as mean ± S.E.M. Statistical significance was tested using an unpaired Student’s *t* test. For experiments using multiple variables, statistical significance was assessed via a two-way ANOVA. A significant difference between two conditions was recorded for * *p* < 0.05, ** *p* < 0.01, *** *p* < 0.001 and **** *p* < 0.0001. 

## 5. Conclusions

By developing a 2D and 3D co-culture system of NSCLC, our results demonstrated that drug-sensitive clones exerted a suppressive effect on the growth of drug-resistant ones. We also revealed, for the first time, that this mechanism could involve metabolic cell activity that depends on lactate dehydrogenase. Moreover, using mathematical modeling of clone expansion and response to treatment, our study highlighted that a metronomic schedule better managed intratumor heterogeneity, while achieving a long-term reduction of tumor volume. Altogether, our study allows us to gain insights into the mechanism by which clones impact each other and could open new therapeutic avenues to manage intratumor heterogeneity in NSCLC.

## Figures and Tables

**Figure 1 cancers-13-02239-f001:**
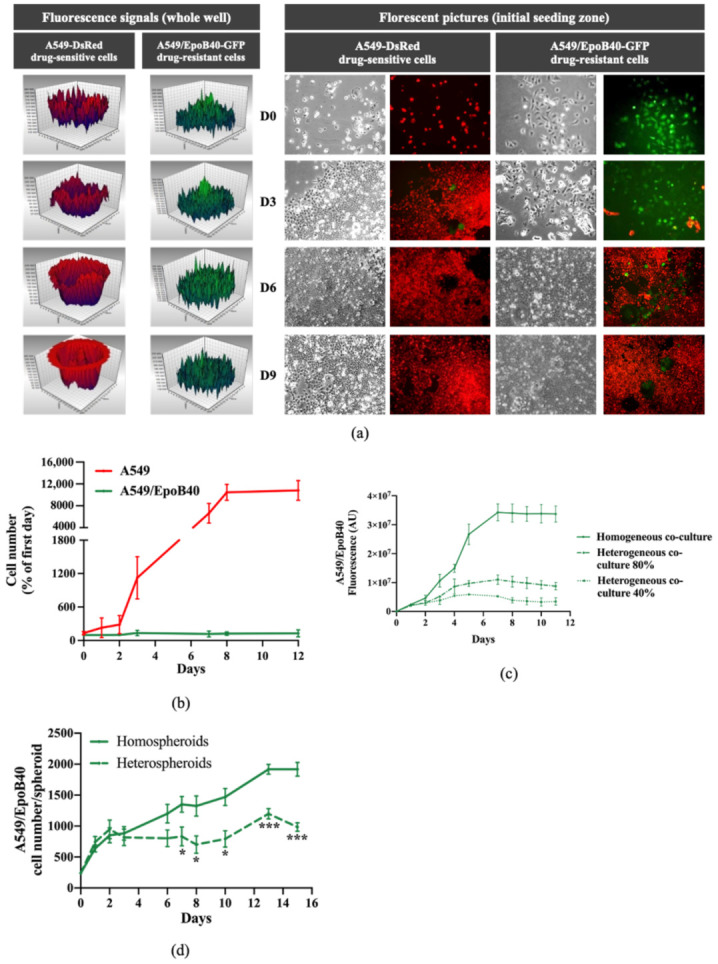
Drug-sensitive A549 clones inhibit the proliferation of the drug-resistant A549/EpoB40 clones. (**a**) Representative fluorescent signal using a well-scanning mode microplate reader (left panel) and microscopic pictures (right panel; magnification factor ×10) of A549-DsRed drug-sensitive and A549/EpoB40-GFP drug-resistant cells over time. (**b**) Cell number of A549 and A549/EpoB40 by recording fluorescent signals (DsRed and GFP) over time. Data were expressed as a percentage of cell number from day 0. (**c**) Fluorescent signal of A549/EpoB40 cells over time in homogeneous or heterogeneous 2D co-culture systems. (**d**) Relative A549/EpoB40 cell numbers in 3D hetero-spheroids and homo-spheroids measured by fluorescent signal over time. Data are shown as mean ± SEM. * *p* < 0.05; *** *p* < 0.001.

**Figure 2 cancers-13-02239-f002:**
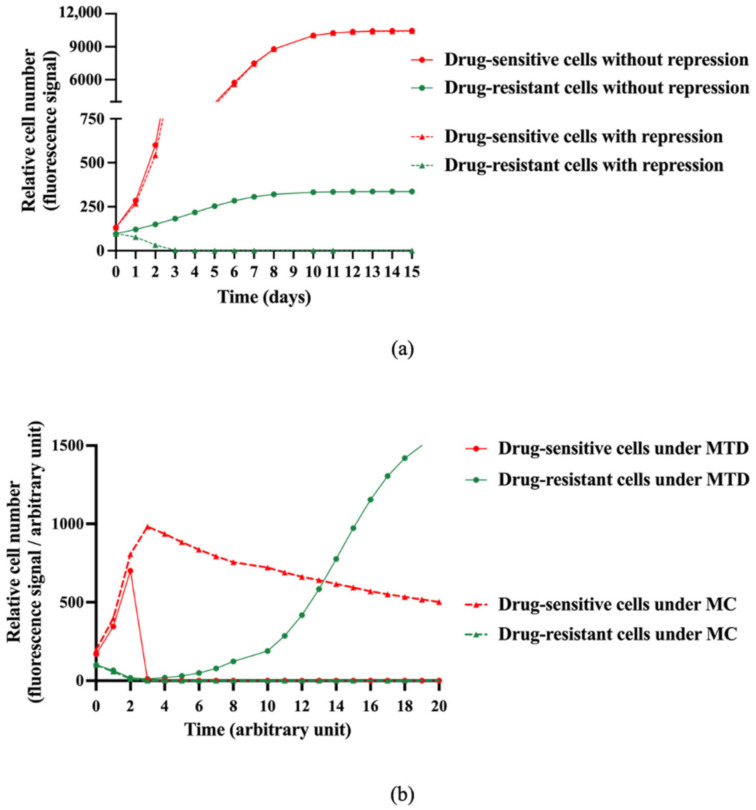
Model-based in silico simulation of the interaction between sensitive and resistant clones with or without treatment. (**a**) Model-based in silico simulations of relative drug-sensitive and drug-resistant cell proliferation over time with or without a repressive effect exerted by the drug-sensitive cells over the drug-resistant ones. (**b**) In silico simulations of relative drug-sensitive and drug-resistant cell proliferation under MTD (Maximum Tolerated Dose) and MC (Metronomic Chemotherapy) schedules and taking into account the suppressive effect of the drug-sensitive cells over the drug-resistant ones (β > 0 in the model).

**Figure 3 cancers-13-02239-f003:**
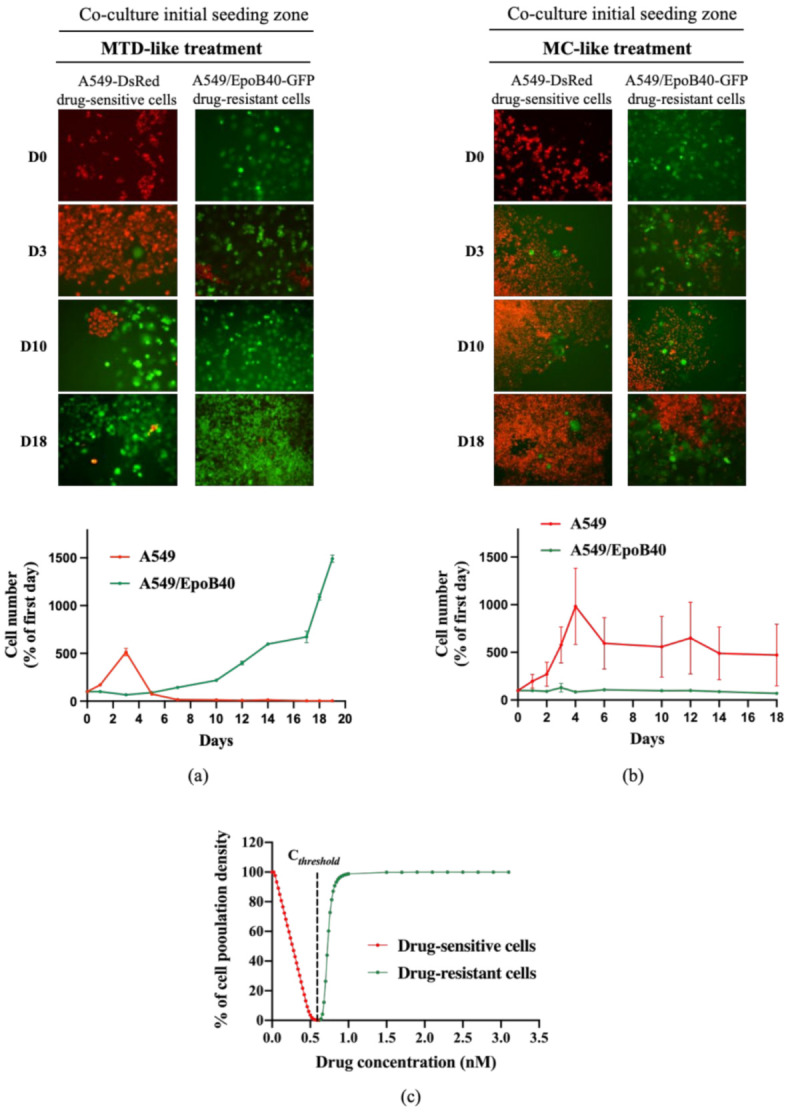
The metronomic schedule prevents the selection of drug-resistant clones in 2D NSCLC co-culture models. (**a**,**b**) Representative fluorescent pictures of A549-DsRed drug-sensitive cells and A549/EpoB40-GFP drug-resistant cells (magnification factor ×10) over time under MTD-like treatment (**a**) or MC-like treatment (**b**). Cell number of A549 and A549/EpoB40 by recording fluorescent signals (DsRed and GFP) over time under MTD- and MC-like treatments is also represented. Data were expressed as a percentage of cell number from day 0. (**c**) Percentage of the carrying capacity of the well occupied by cell populations after a 30-day in silico experiment depending on the drug concentration. Red: the well was mainly filled with drug-sensitive cells at the end of the experiment, and green: the well was mainly filled with drug-resistant cells at the end of the experiment.

**Figure 4 cancers-13-02239-f004:**
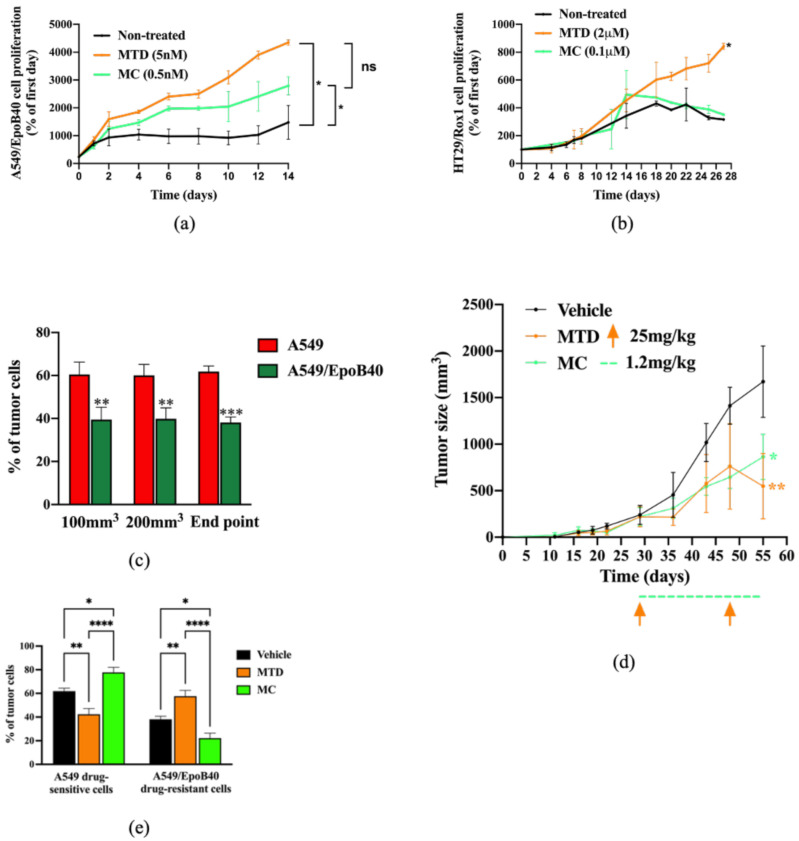
Biological validation of the mathematical model predictions in 3D spheroid and in vivo models. (**a**,**b**) Cell proliferation of A549/EpoB40 (**a**) and HT29/rox1 (**b**) drug-resistant cells by recording fluorescent signals over time under MTD (patupilone 5 nM (**a**) or oxaliplatin 2 μM (**b**)) or MC (patupilone 0.5 nM (**a**) or oxaliplatin 0.1 μM (**b**)) schedules. Data were expressed as a percentage of cell proliferation from day 0. (**c**) Ex vivo tumor cell composition at three different time points in the untreated condition. Fluorescent cell signals were quantified by flow cytometry. (**d**) Tumor sizes were measured over time in the untreated condition as well as under MTD (paclitaxel at 25 mg/kg, once every 3 weeks) and MC treatment (paclitaxel 1.2 mg/kg, daily). Significant differences compared to vehicle were observed. Orange arrows represent MTD-drug administration, while the dotted green line indicates the MC-drug administration period. (**e**) Ex vivo tumor cell composition at the end point in the untreated condition and following MTD-like and MC-like treatment schedules. Fluorescent cell signals were quantified by flow cytometry. Data are expressed as a percentage of drug-sensitive and drug-resistant cell populations and are shown as mean ± SEM. * *p*; ** *p* < 0.01; *** *p* < 0.001; **** *p* < 0.0001.

**Figure 5 cancers-13-02239-f005:**
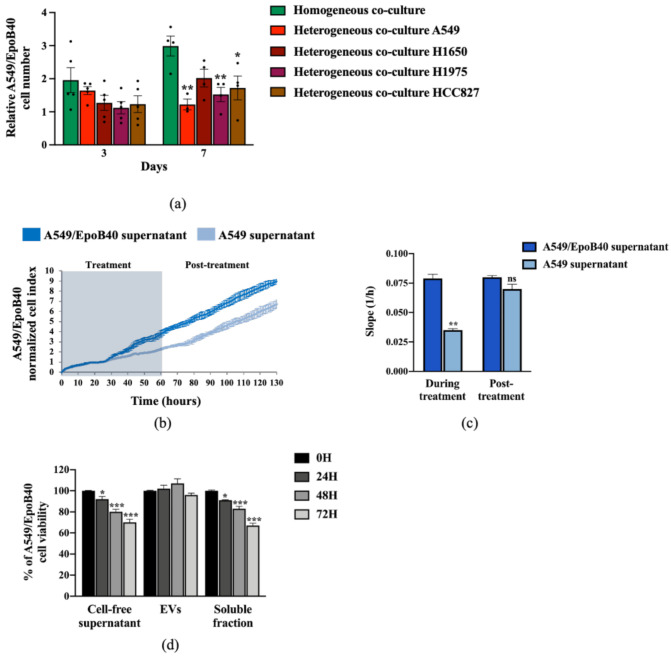
Drug-sensitive A549 clones control the proliferation of drug-resistant A549/EpoB40 clones through a paracrine pathway, independently of exosome secretion. (**a**) Relative quantification of A549/EpoB40 cell number in homogeneous (A549/EpoB40 only) and heterogeneous (A549/EpoB40 with either A549, H1650, H1975 or HCC827) co-culture Transwell^®^. Cells were stained with crystal violet at indicated times. Data were expressed as a ratio of the cell number from day 0. (**b**) A549/EpoB40 cell growth was studied by a real-time impedance-based method. Twenty-four hours after seeding, A549/EpoB40 cells were treated daily with cell-free supernatants from A549 cultures and from A549/EpoB40 cultures that were used as control. Measurements were performed every 15 min. Cell index values were normalized to 24 h, corresponding to the first measurement after starting treatment. The grey-shaded area indicates the duration of treatment. (**c**) Slopes were determined during the period of 48h supernatant treatment and 48h post-supernatant treatment. (**d**) Cell proliferation of A549/EpoB40 cells measured by crystal violet assay after a 0, 24, 48 and 72 h exposition to total cell-free supernatant, extracellular vesicles (EVs) or soluble fraction from A549 cultures. Results were expressed as a percentage of viability at T0. Data are shown as means ± SEM. ns > 0.05; * *p* < 0.05; ** *p* < 0.01; *** *p* < 0.001.

**Figure 6 cancers-13-02239-f006:**
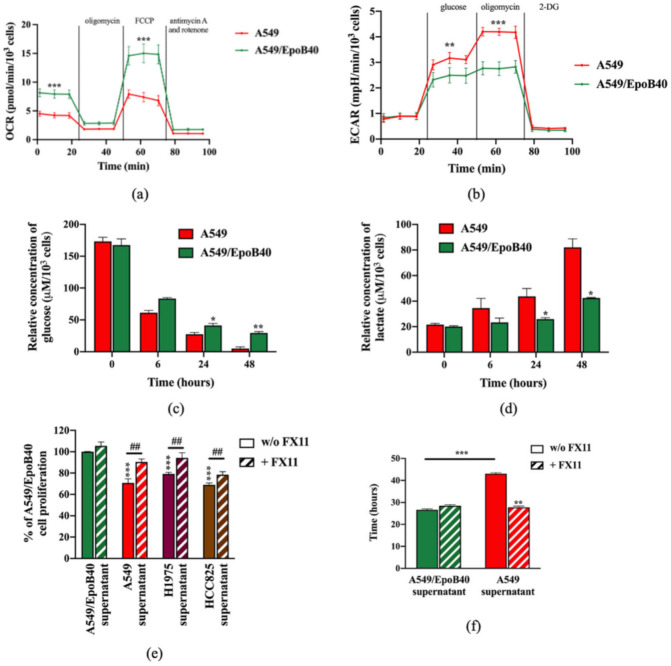
Drug-sensitive A549 clones have a higher glycolytic profile than drug-resistant A549/EpoB40 clones. (**a**,**b**) A549 and A549/EpoB40 cells were analyzed for (**a**) mitochondrial bioenergetics and (**b**) glycolysis using the Seahorse XF technology (OCR: Oxygen Consumption Rate and ECAR: ExtraCellular Acidification Rate). (**c**,**d**) Supernatants from A549 and A549/EpoB40 were collected over time and analyzed for (**c**) glucose consumption and (**d**) lactate production by using the YSI 2900 instrument. (**e**) A549/EpoB40 cell proliferation was measured by crystal violet assay at 72 h after daily exposition to cell-free supernatants from A549, H1975 or HCC825 cultures previously incubated with or without 3 µM of FX11. Supernatants from A549/EpoB40 cultures that were subjected to the same treatment were used as a control. Results were expressed as a percentage of control cell proliferation. (**f**) A549/EpoB40 cell growth was followed by a real-time impedance-based method. A549/EpoB40 cells were exposed daily to cell-free supernatants from A549 and A549/EpoB40 cultures that were previously incubated with or without 3 µM of FX11. Doubling time of A549/EpoB40 cells was calculated in different conditions. Data are shown as means ± SEM. * *p* < 0.05; ** *p* < 0.01; *** *p* < 0.001; ## *p* < 0.01 (* comparison to A549/EpoB40 supernatant w/o FX11 and # comparison between w/o FX11 and FX11 treated conditions).

## Data Availability

The data presented in this study are available in this article (and Appendix A).

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
