# Peer review of "Metronomic Chemotherapy Modulates Clonal Interactions to Prevent Drug Resistance in Non-Small Cell Lung Cancer"

_cancers, 2021, doi:10.3390/cancers13092239_

Round 1
Reviewer 1 Report
See attached report.

Reviewer 2 Report
The manuscript “Metronomic therapy prevents emergence of drug resistance by maintaining the dynamic of intratumor heterogeneity” by Maryna Bondarenko et al, is a good hypothesis, mathematical modeling and experimental article, and could be of interest for cancerologists and general biology readers. However, it needs to be fixed a little bit.
. Title and abstract: neither title or abstract give the lesser idea of what the content of the paper is about. I need to get lo line 102 to understand that point. Of cours I new that had to do something to metronomics in tumours, mathematical modeling and experimental approaches … but nothing specific. Thus, please, be more precise in title and abstract, giving a more accurate info. That is what is to be shown in data bases and both are extremely confusing.
. Be, also, more specific in some keywords: drug resistance (PubMed 550,985 results), cell metabolism (PubMed 2,353,186 results) …
The article is well cast and clear in the three points mentioned hypothesis, mathematical modeling and experimental, Introduction, M&M and discussion could be a little clearer, but, in general are OK.
A final scheme showing a model with your findings will also be advisable.
I saw this manuscript in internet ( DOI:10.1101/2021.01.04.425214) by the end of past January, thus, it calls my attention how I reached it for review 3 months later.
Figures are sharp and clear. Overall a good paper.
Round 2
Reviewer 1 Report
See attached report.
